# Iraqi Parents’ Knowledge, Attitudes, and Practices towards Vaccinating Their Children: A Cross-Sectional Study

**DOI:** 10.3390/vaccines10050820

**Published:** 2022-05-22

**Authors:** Walid Al-Qerem, Anan Jarab, Alaa Hammad, Fawaz Alasmari, Jonathan Ling, Alaa Hussein Alsajri, Shadan Waleed Al-Hishma, Shrouq R. Abu Heshmeh

**Affiliations:** 1Department of Pharmacy, Al-Zaytoonah University of Jordan, Amman 11733, Jordan; alaa.hammad@zuj.edu.jo; 2Department of Clinical Pharmacy, Faculty of Pharmacy, Jordan University of Science and Technology, Irbid 3030, Jordan; asjarab@just.edu.jo (A.J.); srabuheshmeh19@ph.just.edu.jo (S.R.A.H.); 3Department of Pharmacology and Toxicology, College of Pharmacy, King Saud University, Riyadh 11451, Saudi Arabia; ffalasmari@ksu.edu.sa; 4Faculty of Health Sciences and Wellbeing, University of Sunderland, Chester Road, Sunderland SR1 3SD, UK; jonathan.ling@sunderland.ac.uk; 5Specialized Bone Marrow Transplant Center, Medical City Complex, Baghdad 61031, Iraq; alaa94@student.usm.my; 6School of Pharmaceutical Sciences, University of Sains Malaysia, Gelugor 11800, Malaysia; shadenwaleed90@gmail.com

**Keywords:** COVID-19, Iraq, vaccination, parental acceptance, children

## Abstract

The focus of coronavirus disease 2019 (COVID-19) vaccination campaigns has been the adult population, particularly the elderly and those with chronic diseases. However, COVID-19 can also affect children and adolescents. Furthermore, targeting this population can accelerate the attainment of herd immunity. The aim of the current study was to evaluate parental intentions to vaccinate their children and the variables associated with them. An online questionnaire was circulated via generic Iraqi Facebook groups to explore parental intentions regarding the vaccination of their children. Multinomial regression analysis was conducted to evaluate variables associated with parental vaccination acceptance. A total of 491 participants completed the study questionnaire. Only 38.3% of the participants were willing to vaccinate their children against COVID-19, while the rest either refused to vaccinate their children (35.6%) or were unsure whether they would (26.1%). Participants’ perceptions about the effectiveness (OR = 0.726, 95% CI = 0.541–0.975, *p* = 0.033) and safety (OR = 0.435, 95% CI = 0.330–0.574, *p* < 0.0001) of COVID-19 vaccines were significantly associated with parental acceptance of having children vaccinated. Participants who had received or who were planning to receive the COVID-19 vaccine were significantly less likely to reject vaccinating their children (OR = 0.156, 95% CI = 0.063–0.387, *p* < 0.0001). There is high refusal/hesitancy among Iraqi parents to vaccinate their children, which is associated with concerns related to the safety and efficacy of COVID-19 vaccines. More efforts, including educational and awareness campaigns to promote the safety and efficacy of COVID-19 vaccines, should be made to increase parental acceptance of childhood COVID-19 vaccinations in Iraq.

## 1. Introduction

Governments have taken many measures to reduce the risk of infection from coronavirus disease 2019 (COVID-19) as well as the rates of hospitalization and death [1]. These measures have included maintaining social distancing, washing hands, using sanitizers, and, in many countries, imposing curfews [1]. Despite these preventive measures, COVID-19 has continued to spread, causing widespread infections and deaths. According to the World Health Organization (WHO), by the end of March 2022 Iraq had more than two million people infected with COVID-19 and over 24,000 fatalities [2].

Iraq’s Ministry of Health stated in March 2021 that the first shipment of COVID-19 vaccines had arrived [3]. By the end of March 2022, more than 9.5 million Iraqis had received the COVID-19 vaccine, which constitutes more than 23% of the Iraqi population [2]. In November 2021, the Iraqi Ministry of Health announced the opening of more vaccination outlets across the country and that those over 12 years of age were to be vaccinated in order to increase vaccination coverage [4].

Parental vaccine hesitancy is defined as parents’ hesitation, uncertainty, or reluctance to vaccinate their children despite vaccine availability, resulting in a drop in vaccine uptake among children [5]. Many factors may influence parents’ intentions to vaccinate their children, including misinformation and conspiracy theories concerning COVID-19 vaccinations, which foster a negative attitude towards vaccines [6]. Several studies have investigated parents’ perspectives with regard to immunizing their children against COVID-19 and have produced inconsistent results. A survey conducted in Italy revealed a 40% rate of vaccine hesitancy [7]. According to another study conducted in Brazil, only 2.8% of parents were hesitant about vaccinating their children [8], whereas a study conducted in lower- and middle-income countries reported that parental hesitancy was less than 10% for highly effective vaccines [9]. In the Middle East region, parental acceptance ranged from 28.9% in Turkey [10] to 75% in Saudi Arabia [11].

Consequently, it is critical to assess COVID-19 vaccination acceptance and collect data on immunization views and attitudes among parents in different parts of the world. This will allow public health messages to be tailored according to target populations. The purpose of this study is to measure the extent of COVID-19 vaccination hesitancy and to evaluate variables associated with it among the Iraqi population.

A previous review showed substantial variation in COVID-19 vaccine acceptance rates among adults. The overall acceptance rates among general populations in East and Southeast Asia were relatively high, including acceptance rates of over 90% reported in Indonesia, Malaysia, and in one Chinese study [12,13]. In Western/Central Europe and North America, the highest rates were reported in Canada (91%) and Norway (89%), while the lowest rates were reported in Cyprus and Portugal (35%) [14]. In Latin America, COVID-19 vaccine acceptance rates ranged from 88% in Mexico to 43% in Haiti, and in Africa, the highest acceptance rate was in Ethiopia (92%), while the lowest was in Zimbabwe (50%) [14]. A later survey from Shenzhen, China, by Zhang et al., which asked factory employees’ parents/guardians about the acceptability of COVID-19 vaccination for their children, found a lower acceptance rate of 72.5% [15]. Similarly, an online survey of Australian parents found a 75.8% approval rate, down from 85.8% among people in Australia who were surveyed in April 2020 [16,17].

The Middle East has some of the lowest COVID-19 vaccination adoption rates in the world. Kuwait (23.6%) had the lowest acceptance rate, followed by Jordan (36.8%), Saudi Arabia (64.7%), and Turkey (66.0%) [18,19,20]. Low vaccination rates can be attributed to the region’s extensive embrace of conspiratorial views, which has resulted in a negative attitude toward immunization [21,22]. However, Israel had the highest vaccination acceptance rate (75.0%); nevertheless, this rate was substantially lower (61.1%) among nurses surveyed in the same study [23]. Moreover, parental acceptance of COVID-19 vaccination was also low in the neighboring country of Jordan (30.2%) [24]. COVID-19 vaccine acceptance could be a stumbling block in worldwide attempts to contain the present pandemic’s harmful health and socioeconomic consequences. Thus, different studies are needed around the world to improve understanding of the factors that will influence vaccine acceptance. The aim of the current study was to evaluate parents’ acceptance of vaccination for their children and to evaluate associated factors among the Iraqi population.

## 2. Materials and Methods

### 2.1. Study Design and Subjects

This was an online cross-sectional study. The questionnaire was created in Google Forms and the link was distributed among various generic Iraqi Facebook groups in posts that included a brief description of the study aims. The study enrolled any Iraqi parents with children who had not been immunized against COVID-19. To ensure that all study participants met the inclusion criteria, potential respondents were asked to provide information about a place of residence and whether they had children (under the age of 18) who were unvaccinated against COVID-19 prior to the questionnaire. Those who did not meet the inclusion criteria were automatically excluded from completing the survey. Children who were born prematurely, children who take immunosuppressive drugs, and children who suffer from cancers, asthma, sickle cell anemia, thalassemia, allergies, diabetes, liver disease, and circulatory diseases were considered to have a high risk for developing COVID-19 infection. Data were collected from September 2021 through February 2022.

### 2.2. Ethical Approval

The study was conducted according to the ethical principles of the Declaration of Helsinki [25]. Ethical approval was obtained from the ethical committee of the Medical City Complex (IRB: 2021/04/02). Consent forms were included at the beginning of the questionnaire.

### 2.3. Sample Size Calculation

The Eakrejcie and Morgan method for nonspecific population size was used to determine the required sample size [26]. The required number of participants for this study was calculated at 95% confidence interval and 5% confidence level and was equal to 385 participants. The Rule of Events per Variable criterion (EPV) ≥ 10 [27] was used to determine the sample required to carry out a logistic regression. As the model included 11 variables, the smallest group had to number at least 110. In this study, the smallest group included 128 participants (the “Not sure” group); therefore, the minimum required sample size condition was met.

### 2.4. Study Instruments

The study utilized a previously validated questionnaire used to measure parents’ willingness to vaccinate their children against COVID-19 in Jordan [24]. Although the questionnaire was previously validated, a pilot study was conducted to perform face validity and intelligibility in order to make certain of the questionnaire’s clarity for Iraqi participants. The pilot study included 30 randomly selected participants who were enrolled by approaching attendees at three pharmacies in different locations in Iraq and who confirmed the questionnaire’s clarity (the main sample characteristics are included in Appendix A). The results from the pilot study were not included in the final data set.

The beginning of the questionnaire included an introduction explaining the objectives of the study and that continuing to fill out the questionnaire indicated consent to participate in the study. The questionnaire was divided into five sections. The demographic part of the questionnaire included variables such as age, gender, marital status, educational qualifications, household income, number of children (including gender and age), and whether the participant was vaccinated or intended to get vaccinated against COVID-19.

Further sections of the questionnaire asked about the health status of the participants and their children, participants’ experience of COVID-19, including whether they, their children, or their relatives have been previously infected. Participants also provided data on their perceptions of the risk of COVID-19 for them and for their children, on their knowledge about COVID-19, including symptoms, means of protection from contracting the disease, mechanisms of transmission, the extent to which the participants had adopted preventive measures, their perceptions of the safety and effectiveness of available vaccines, and, lastly, on their willingness to vaccinate their children against COVID-19. The perceptions of the participants regarding the safety and efficacy of the vaccine and the seriousness of COVID-19 were measured using a five-point Likert-type response scale, where 1 indicated low and 5 indicated high levels of perceived vaccine effectiveness and safety and disease risk. Parents who stated their unwillingness to have their children vaccinated were also asked further questions in order to understand the reasons for their decision. Knowledge and practice scores were calculated for each participant. Knowledge was calculated with 19 points (Appendix A). Practice scores were calculated with 25 points based on five questions that evaluated adherence to protective practices against COVID-19. Responses were on a 5-point Likert scale, which ranged from 1 (never) to 5 (all the time) (Appendix A).

Participants were divided into a high-level group and a low-level group based on knowledge and practice median scores, the high-level group including participants scoring higher than the median.

### 2.5. Statistical Analysis

The data were analyzed using SPSS version 26 [28]. Categorical variables were presented as frequencies and percentages, and continuous variables were presented as means and standard deviations (SD). To evaluate the internal consistency of the questionnaire, Cronbach’s alphas were computed for the protective practices and knowledge scales. The minimum acceptable Cronbach’s alpha was 0.7 for the attitude score and 0.5 for the knowledge score, as a lower Cronbach’s alpha is acceptable for binary data [29]. Kruskal–Wallis and chi-square analyses were used to identify variables associated with participants’ intentions to vaccinate their children, and significant variables (*p* < 0.05) were included in the multinominal logistic regression model. The Nagelkerke pseudo-R-square value produced by the multinomial model is reported.

## 3. Results

A total of 491 individuals participated in the study. Most of the participants were female (59.3%), married (78.6%), had a monthly income of <$500 (42.6%), had completed university education (58.4%), non-smokers (88.6%), and did not suffer from any chronic diseases (88.6%). Most of the participants reported having no children with a chronic disease (93.7%) or who were born prematurely (90.2%). Almost half of the participants were in the age group of 30–39 (47%) and had children of both sexes (48.3%); a third of them had two children (34.8%). The majority of children were aged between 7 and 12 years (42.8%), and only 9.6% of them were under the age of one year. Demographic characteristics of the study participants are presented in Table 1.

Participants’ adherence to protective practices against COVID-19 and their knowledge about COVID-19 and its vaccine were classified into low and high based on median scores. The medians for the practice and knowledge scores were 20 (quartiles: 17–22) and 16 (quartiles: 13–18) respectively. Those who scored above the median were included in the high-level group, while those whose scores were below the median were included in the low-level group. Cronbach’s alphas for adherence to practice measures and knowledge about COVID-19 were 0.72 and 0.85, respectively.

Results showed that only 38.3% of the participants indicated a willingness to vaccinate their children against COVID-19, while the rest were either unwilling (35.6%) or unsure about vaccinating their children (26.1%). As shown in Table 2, univariate analysis showed that participants’ intentions to vaccinate their children against COVID-19 were significantly affected by education (*p* = 0.022), whether their children had had COVID-19 (*p* < 0.0001), whether they had been infected with COVID-19 themselves (*p* = 0.021), their anticipation of the likelihood of a COVID-19 infection in their children during the next 6 months (*p* = 0.009), their knowledge about COVID-19 and its vaccines (*p* = 0.004), their adherence to protective practices against COVID-19 (*p* = 0.006), and whether they had received or planned to receive the COVID-19 vaccine (*p* < 0.0001).

The Nagelkerke pseudo-R-square value produced by the multinomial model was 0.44. The multinomial regression showed that each unit increase in participants’ perceptions of the effectiveness of COVID-19 vaccines for their children decreased the odds of responding “No” vs. “Yes” (OR = 0.726, 95% CI = 0.541–0.975, *p* = 0.033). Each unit increase in the perceived safety of the COVID-19 vaccine for children decreased the odds of responding “No” or “Not sure” vs. “Yes” (OR = 0.435, 95% CI = 0.330–0.574, *p* < 0.0001 and OR = 0.587, 95% CI = 0.451–0.763, *p* < 0.0001, respectively). Participants who answered “Maybe” to the question “Have any of your children ever been infected with COVID-19?” were more likely to respond “Not Sure” vs. “Yes” (OR = 3.127, 95% CI = 1.317–7.422, *p* = 0.010) compared to those who answered “Yes”. Furthermore, participants who had received or who were planning to receive a vaccination against COVID-19 were significantly less likely to respond with “No” or “Not Sure” vs. “Yes” (OR = 0.156, 95% CI = 0.063–0.387, *p* < 0.0001 and OR = 0.238, 95% CI = 0.096–0.592, *p* = 0.002, respectively) compared to those who had not received the vaccine. Results of the multivariate analysis of the variables associated with a participant’s responding “Not Sure” or “No” to the question about their intention to vaccinate their children are shown in Table 3.

As shown in Table 4, the most frequently recognized reasons for participants’ hesitancy or refusal to vaccinate their children were: “The vaccine has not been adequately tested on children” (89.8%), followed by “I don’t want a vaccine I know nothing about. I’ll make my decision if/when one becomes available” (82.8%) and “I am scared to put a foreign object in my children’s body” (77.9%). “Vaccination causes autism” was the least reported reason.

## 4. Discussion

At the beginning of the COVID-19 pandemic, the adult population was the main target of vaccination campaigns. This was both because children rarely become severely ill from COVID-19 as well as a lack of evidence supporting the safety and efficacy of using COVID-19 vaccines in children. Nevertheless, in the USA, COVID-19 infection is the seventh highest cause of death for children aged 5–11 years and even higher than that for children aged 12–18 years [30]. COVID-19 vaccines have been shown to be safe and effective at preventing symptomatic COVID-19 in children. Therefore, approval for vaccinating children aged 12–17 years against COVID-19 has been given, and the schedule has been accelerated to include those aged 5–11 years in the immunization process during the second half of 2021 [31]. However, parents are often hesitant about getting their children vaccinated against COVID-19 [32], which could be a barrier to the immunization process and the achievement of herd immunity. Findings from the present study provide insights for the development of healthcare strategies to increase parental willingness to have their children vaccinated against COVID-19 and overcome the barriers associated with parental hesitancy or refusal of vaccinations for their children.

The current study found low parental COVID-19 vaccine acceptance for children, with only 38.3% of parents reporting willingness to vaccinate their children against COVID-19. This finding was in line with those of studies conducted across different countries, including Jordan [24], Turkey [33], Australia [34], and the USA [35], and lower than figures reported in studies conducted in the UK [36], the USA [32], Saudi Arabia [11], Japan [37], Korea [38], China [39], Brazil [8], Italy [7], and Qatar [40].

Consistent with the findings of an earlier study [24], our results revealed that stronger perceptions about the effectiveness of a COVID-19 vaccine for children significantly reduced parental vaccination refusal. Perceptions about COVID-19 vaccine effectiveness have also been associated with parents’ acceptance of vaccination against COVID-19 for their children in previous studies [10,11] and with COVID-19 vaccination acceptance in other population groups, including adults [41,42,43], university students [44], and healthcare workers [45]. This is also one of the barriers to COVID-19 vaccine acceptance in general populations and other subgroups [42]. A Chinese study reported that parental exposure to positive information about COVID-19 vaccines on social media was associated with higher parental acceptance of COVID-19 vaccinations for their children, which may be related to the positive role of this type of information in increasing parents’ confidence in COVID-19 vaccines and subsequently increasing their desire to vaccinate their children [15]. Similarly, a study conducted with older people found that participants who reported social media as the main source of information about COVID-19 vaccines were more willing to receive a COVID-19 vaccine [46]. Another study found that higher parental trust in COVID-19 vaccines was associated with lower hesitancy regarding child vaccinations [47]. Therefore, it is essential to increase parental knowledge about COVID-19 vaccines, as this can significantly increase parents’ willingness to vaccinate their children.

The perceived safety of COVID-19 vaccines affected parents’ decisions about vaccinating their children in the current study, which was consistent with what has been reported in Korean [38], and Jordanian populations [24]. Earlier studies found that hesitancy was largely due to safety concerns [36,45]. Although the majority of the parents who participated in a Brazilian study were willing to accept COVID-19 immunization for their children, more than half were concerned about serious side effects of COVID-19 vaccines and one third were concerned about their safety [8]. Furthermore, a literature review found that the main reasons for parents’ hesitancy about vaccinating their children against COVID-19 were fears of side effects and safety concerns [48]. Moreover, concerns about the safety profile of COVID-19 vaccines has been identified as a factor in vaccination hesitancy among the general population [49], the elderly [50], and pharmacy students [51]. Therefore, public health authorities must address misinformation about the safety of COVID-19 vaccines and provide supportive evidence related to vaccine safety, this having been recognized as the factor most relevant to convincing reluctant parents to have their children receive a COVID-19 vaccination [39].

We found that parents who were skeptical about the likelihood of their children being infected with COVID-19 were less sure about vaccinating their children against COVID-19 than parents who knew that their children had encountered the infection. In comparison, participants who were doubtful about their children being infected with COVID-19 were less likely to refuse to vaccinate their children compared to those who denied that their children had been infected [24]. This finding could be explained by the fact that parents who had children infected with COVID-19 would like to protect them from being re-infected. This was reported as an effective persuading factor to get parents to vaccinate their children against COVID-19. On the other hand, parents who did not have infected children or who were doubtful about the danger of COVID-19 felt that the risk of infection was lower than those whose children had suffered from a COVID-19 infection.

Those parents in the current study who were vaccinated or who were planning to receive a COVID-19 vaccine reported greater willingness to vaccinate their children than those who did not intend to be vaccinated. Earlier work has also reported that parents’ willingness to vaccinate themselves is significantly associated with intention to vaccinate their children [35,44]. Likewise, a Jordanian study found a lower acceptance for child vaccinations against COVID-19 among parents who had not received a COVID-19 vaccine and those who were not planning to receive the vaccine themselves [24]. Another factor significantly associated with parental attitudes towards COVID-19 vaccination was education, which has also been reported to impact parental perceptions about other child vaccinations [9] in addition to COVID-19 [52]. Education level has also affected COVID-19 vaccine acceptance among different populations, such as the elderly [46,53] and community adults [54]. This emphasizes the importance of targeting low-education groups in different awareness programs.

The majority of the parents in the present study were concerned about the novelty and the rapid development of the COVID-19 vaccine which they felt might not have allowed for adequate testing to confirm its safety and efficacy for use in children. Similarly, a Jordanian study reported that the majority of the study participants were hesitant about using a COVID-19 vaccine on their children due to its newness and the lack of adequate information about it [24]. Most of the participants surveyed in a British study reported their main reason for not accepting COVID-19 vaccination in children was that the vaccine was new, rushed, and had insufficient evidence to support its safety and efficacy (68%) [21]. Such results have been replicated elsewhere [33].

We found that 13.2% of the participants indicated that “Vaccines causes autism” as the reason behind their refusal or hesitancy toward childhood vaccinations against COVID-19. This is consistent with the findings reported in an earlier study [24] and higher than the figure reported in another study [11]. These findings should shed light on the importance of increasing public awareness about the safety and efficacy of COVID-19 vaccines in children through the provision of evidence-based data that would increase parents’ trust in COVID-19 vaccines and eventually promote their willingness to vaccinate their children—not only against COVID-19, but other diseases.

It is worth noting that the mandatory implementation of a green pass—a certificate issued to individuals who are vaccinated, who were infected by COVID-19 and recovered, or those with a negative rapid antigen test performed within the past 48 h or a negative molecular test performed within the last 72 h [55]—could be associated with lower COVID-19 vaccine acceptance [46]. This mandatory policy could be regarded by some as a threat to civil liberties, resulting in a drop in vaccine acceptability [55]. Therefore, future research should investigate the relationship between compulsory measures and parental willingness to vaccinating their children against COVID-19.

### Study Limitations

The current study used a self-administered questionnaire which may have increased the chance of selection and/or recall biases. Nevertheless, the statistically sufficient sample size included in this study and the widespread use of internet in Iraq that reached 60% of the total population by 2019 [56] could have minimized the bias effect. This is evident when comparing some of the sample characteristics when possible with the general Iraqi population; for example, the positively skewed sample distribution in age and household average monthly income is similar to the general Iraqi population [57,58]. Despite these limitations, web-based studies are a cost-effective method that can produce a representative sample, provide a safe environment for the respondents to answer questions honestly and accurately, and remove interviewer bias [59,60]. Unfortunately, duplicated submissions can only be prevented in Google Forms by requesting respondents’ email addresses, which may jeopardize the anonymity of the questionnaire. Therefore, the current study may have included duplicated submissions; however, visual, and computerized (using SPSS tools) were utilized to detect any duplicated responses.

## 5. Conclusions

This research indicates that there is a high rate of hesitancy among Iraqi parents to vaccinate their children. This hesitancy was related to perceptions of the safety and efficacy of COVID-19 vaccines. Targeted campaigns should be developed to increase parental acceptance of vaccinations against COVID-19 for their children.

## Figures and Tables

**Table 1 vaccines-10-00820-t001:** Sociodemographic characteristics of the participants and their children.

Characteristics		Frequency (%) (*n* = 491)
Sex	Male	200 (40.7)
	Female	291 (59.3)
Age	18–29 years	168 (34.2)
	30–39 years	231 (47.0)
	40 years or above	92 (18.8)
Marital status	Not currently married	105 (21.4)
	Currently married	386 (78.6)
Educational level	High school or less	136 (27.7)
	Diploma	32 (6.5)
	University student	36 (7.3)
	Bachelor’s degree	177 (36.0)
	Postgraduate	110 (22.4)
Household average monthly income ($)	Less than $500	209 (42.6)
	$500–$1000	191 (38.9)
	More than $1000	91 (18.5)
Working/studying in a medical field	Yes	256 (52.1)
	No	235 (47.9)
Smoking	No	435 (88.6)
	Yes	56 (11.4)
Having a chronic disease	No	435 (88.6)
	Yes	56 (11.4)
Number of children	1	119 (24.2)
	2	171 (34.8)
	3	90 (18.3)
	4 or more	111 (22.6)
Children’s sex	Boys	142 (28.9)
	Girls	112 (22.8)
	Boys and girls	237 (48.3)
Children’s age	Under one year	47 (9.6)
	1–3 years	190 (38.7)
	4–6 years	206 (42)
	7–12 years	210 (42.8)
	13–17 years	123 (25.1)
Having a child with a chronic disease	No	460 (93.7)
	Yes	31 (6.3)
Having a child born prematurely	No	443 (90.2)
	Yes	48 (9.8)

**Table 2 vaccines-10-00820-t002:** Bivariate analysis of intentions to vaccinate children by different sample characteristics presented as means (SD) or frequencies (percentages).

	Are You Willing to Vaccinate Your Children against COVID-19?			*p*-Value
	No (N = 175, 35.6%)	Not Sure (N = 128, 26.1%)	Yes (N = 188, 38.3%)	
How many children do you have?				0.074
One	41 (23.4)	35 (27.3)	43 (22.9)	
Two	54 (30.9)	38 (29.7)	79 (42.0)	
Three	37 (21.1)	19 (14.8)	34 (18.1)	
Four or more children	43 (24.6)	36 (28.1)	32 (17.0)	
Children’s age				
Less than one year	18 (10.3%)	12 (9.4%)	17 (9.0%)	0.919
1–3 years	72 (41.1%)	51 (39.8%)	67 (35.6%)	0.534
4–6 years	77 (44.0%)	45 (35.2%)	84 (44.7%)	0.192
7–12 years	77 (44.0%)	54 (42.2%)	79 (42.0%)	0.919
13–17 years	47 (26.9%)	32 (25.0%)	44 (23.4%)	0.750
Participant age				0.561
18–29	67 (38.3)	44 (34.4)	57 (30.3)	
30–39	75 (42.9%)	60 (46.9%)	96 (51.1%)	
40 or more	33 (18.9%)	24 (18.8%)	35 (18.6%)	
Children’s sex				0.076
Girls	35 (20.0)	39(30.5)	38 (20.2)	
Boys	46 (26.3)	33 (25.8)	63 (33.5)	
Both (girls and boys)	94 (53.7)	56 (43.8)	87 (46.3)	
Marital status				0.427
Not currently married	43 (24.6)	26 (20.3)	36 (19.1)	
Currently married	132 (75.4)	102 (79.7)	152 (80.9)	
Sex				0.997
Female	104 (59.4)	76 (59.4)	111 (59.3)	
Male	71 (40.6)	52 (40.6)	77 (41.0)	
Smoking				0.836
No	157 (89.7)	113 (88.3)	165 (87.8)	
Yes	18 (10.3)	15 (11.7)	23 (12.2)	
Education level				0.022
High school or less	51 (29.1)	44 (34.4)	41 (21.8)	
University student	19 (10.9)	8 (6.2)	9 (4.8)	
Diploma	10 (5.7)	11 (8.6)	11 (5.9)	
Bachelor’s degree	52 (29.7)	42 (32.8)	83 (44.1)	
Postgraduate	43 (24.6)	23 (18.0)	44 (23.4)	
Household average monthly income				0.065
Less than $500	72 (41.1)	60 (46.9)	77 (41.0)	
$500–1000	80 (45.7)	43 (33.6)	68 (36.2)	
More than $1000	23 (13.1)	25 (19.5)	43 (22.9)	
Are you working/studying in a medical field?				0.729
No	81 (34.5)	65 (27.7)	89 (37.9)	
Yes	94 (36.7)	63 (24.6)	99 (38.7)	
Do you have a chronic disease?				0.978
No	155 (35.6)	114 (26.2)	166 (38.2)	
Yes	20 (35.7)	14 (25.0)	22 (39.3)	
Do you know somebody close to you who was infected with COVID-19?				0.058
No	49 (28.0)	32 (25.0)	49 (26.1)	
Not sure	27 (15.4)	25 (19.5)	16 (8.5)	
Yes	99 (56.6)	71 (55.5)	123 (65.4)	
Have any of your children ever been infected with COVID-19?				<0.0001
No	119 (68.0)	74 (57.8)	111 (59.0)	
Not sure	20 (11.4)	39 (30.5)	30 (16.0)	
Yes	36 (20.6)	15 (11.7)	47 (25.0)	
Have you ever been infected with COVID-19?				0.021
No	76 (43.4)	53 (41.4)	70 (37.2)	
Not sure	19 (10.9)	29 (22.7)	25 (13.3)	
Yes	80 (45.7)	46 (35.9)	93 (49.5)	
In your opinion, what is the likelihood that your children will be infected with COVID-19 during the next 6 months?				0.009
I do not think that my child will be infected	90 (51.4)	51 (39.8)	70 (37.2)	
I think that my child will be infected and their symptoms will be mild	78 (44.6)	69 (53.9)	96 (51.1)	
I think that my child will be infected and their symptoms will be severe	7 (4.0)	8 (6.2)	22 (11.7)	
Child risk for COVID-19				0.393
Low	150 (35.0)	116 (27.1)	162 (37.9)	
High	25 (39.7%)	12 (19.0)	26 (41.3)	
Knowledge level				0.004
Low	129 (73.7)	73 (57.0)	102 (54.3)	
High	46 (26.3)	55 (43.0)	86 (45.7)	
Practice level				0.006
Low	116 (66.3)	84 (65.6)	97 (51.6)	
High	59 (33.7)	44 (34.4)	91 (48.4)	
Have you received or are you planning to receive a COVID-19 vaccine?				<0.0001
No	67 (61.5)	34 (31.2)	8 (7.3)	
Yes	108 (28.3)	94 (24.6)	180 (47.1)	

**Table 3 vaccines-10-00820-t003:** Multivariate predictors of a participant’s responding “Not sure” or “No” to the question about their intention to vaccinate their children.

Characteristics	Intent to Vaccinate No vs. Yes OR (95% CI)	Intent to Vaccinate Not Sure vs. Yes OR (95% CI)
Estimate seriousness of COVID-19 on child	0.911 (0.699–1.186)	0.840 (0.668–1.055)
Estimate of the seriousness of COVID-19 on participants	0.915 (0.720–1.163)	1.037 (0.805–1.335)
How effective is the use of a COVID-19 vaccine for your children?	0.726 (0.541–0.975) *	0.874 (0.659–1.160)
In your opinion, how safe is the use of a COVID-19 vaccine for your children?	0.435 (0.330–0.574) **	0.587 (0.451–0.763) **
Education level		
High school or less	0.547 (0.230–1.302)	1.511 (0.649–3.520)
University student	2.306 (0.737–7.218)	2.079 (0.618–6.991)
Diploma	0.780 (0.234–2.593)	1.785 (0.585–5.447)
Bachelor’s degree	0.662 (0.335–1.307)	0.891 (0.447–1.775)
Postgraduate	**Reference**	
Have you ever been infected with COVID-19?		
No	0.982 (0.504–1.914)	1.422 (0.732–2.763)
Maybe	0.842 (0.345–2.052)	1.739 (0.798–3.790)
Yes	**Reference**	
Have any of your children ever been infected with COVID-19?		
No	1.412 (0.645–3.090)	1.696 (0.739–3.893)
Maybe	1.045 (0.422–2.589)	3.127 (1.317–7.422) **
Yes	**Reference**	
In your opinion, what is the likelihood that your children will be infected with COVID-19 during the next 6 months?		
I think that my child will be infected and their symptoms will be severe	0.383 (0.124–1.179)	0.711 (0.257–1.968)
I think that my child will be infected and their symptoms will be mild	0.755 (0.427–1.334)	1.097 (0.623–1.932)
I do not think that my child will be infected	**Reference**	
Knowledge level		
Low	1.12 (0.626–1.976)	0.624 (0.354–1.102)
High	**Reference**	
Practice level		
Low	1.508 (0.867–2.623)	1.344 (0.785–2.302)
High	**Reference**	
Have you received or are planning to receive a COVID-19 vaccine?		
Yes	0.156 (0.063–0.387) **	0.238 (0.096–0.592) **
No	**Reference**	

Notes: * Significant at *p* < 0.05. ** Significant at *p* < 0.01.

**Table 4 vaccines-10-00820-t004:** Reasons for a participant’s responding “No” or “Not sure” to the question about their intention to vaccinate their children.

Reasons	Total N (%)
Concern about the vaccine	
The vaccine has not been adequately tested on children	272 (89.8)
Vaccines cause autism	40 (13.2)
I don’t think that I can afford the vaccine	60 (19.8)
Vaccination may cause infected infectious diseases	168 (55.4)
Need additional information	
I don’t want a vaccine I know nothing about. I’ll make my decision if/when one becomes available	251 (82.8)
Attitudes	
I don’t think my child is at risk of contracting the virus	131 (43.2)
I am scared to put a foreign object into my child’s body	236 (77.9)
Lack of trust	
If government agencies recommend the vaccination, I will not give it to my child	161 (53.1)
There is no way I trust big pharmaceutical companies	161 (53.1)
I believe that this virus was developed by governments and I won’t give my children any vaccine	134 (44.2)

## Data Availability

The data presented in this study are openly available in Zenodo at 10.5281/zenodo.6426139.

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
