# Peer review of "Iraqi Parents’ Knowledge, Attitudes, and Practices towards Vaccinating Their Children: A Cross-Sectional Study"

_vaccines, 2022, doi:10.3390/vaccines10050820_

Round 1
Reviewer 1 Report
Iraqi parents’ knowledge, attitudes, and practices towards vaccinating their children: A cross-sectional study
Comment:
The aim of the current study was to evaluate parental intention to vaccinate their children and the variables associated with it. The study is an extension of the work conducted by the same group of researcher https://www.dovepress.com/covid-19-vaccination-acceptance-and-its-associated-factors-among-the-i-peer-reviewed-fulltext-article-PPA
Comment:
The manuscript is well written and carefully tabulated results with reliable statistical analysis. I have some minor comments that the authors could consider:
ABSTRACTS:
Only 38.3% of the participants were willing to vac-cinate their children against COVID-19, while the rest either refused to vaccinate their children (35.6%) or were unsure whether they would (26.1%). Participants’ perception about the effectiveness (OR= 0.726, 95% CI=0.541-0.975, p=0.033) and safety (OR= 0.435, 95% CI=0.330-0.574, p<0.0001) of COVID-19 vaccines were significantly associated with parental refusal for providing the vaccine for their children. Participants who had received or who were planning to take COVID-19 vaccine were sig-nificantly less likely to reject vaccinating their children (OR= 0.156, 95% CI=0.063-0.387, p<0.0001).
Comment:
These sentences could be rephrased to align the reported OR values: Participants’ perception about the effectiveness (OR= 0.726, 95% CI=0.541-0.975, p=0.033) and safety (OR= 0.435, 95% CI=0.330-0.574, p<0.0001) of COVID-19 vaccines were significantly associated with parental refusal for providing the vaccine for their children
Comment:
Conclusion: More efforts should be made to increase parental acceptance of childhood COVID-19 vaccinations in Iraq. -this is rather general and generic statement
INTRODUCTION
Comment:
The aim should be explicitly statement in this section.
Materials and Methods
This cross-sectional study used an online survey distributed as a Google Forms link. The questionnaire was distributed to different regions of Iraq using public social networking sites including Facebook, WhatsApp, Viber, Instagram and Telegram. The questionnaire was distributed between May-July 2021.
Comment: any measure taken to prevent duplicated submission from the same person via different channels?
Statistical Analysis
To evaluate internal consistency of the questionnaire, Cronbach’s alphas were computed for the protective practices and knowledge scales. The minimum acceptable Cronbach’s alpha was 0.7.
Comment:
Cronbach’s alpha was used to evaluate the internal consistency of the items used to compute the knowledge and practice scores. The minimum acceptable alpha for the practice score and knowledge estimated using the binary data should be included.
Author Response
We would like to thank the reviewer for his comments that significantly improved the quality of the manuscript
Please find below our responses
Reviewer comments:
Comment:
The aim of the current study was to evaluate parental intention to vaccinate their children and the variables associated with it. The study is an extension of the work conducted by the same group of researcher https://www.dovepress.com/covid-19-vaccination-acceptance-and-its-associated-factors-among-the-i-peer-reviewed-fulltext-article-PPA
-Thank you for your comment. This was mentioned in the manuscript in the method section: “The study utilized a previously validated questionnaire used to measure parents' willingness to vaccinate their children against the COVID-19 in Jordan [24]”
Comment:
The manuscript is well written and carefully tabulated results with reliable statistical analysis. I have some minor comments that the authors could consider
ABSTRACTS:
Only 38.3% of the participants were willing to vac-cinate their children against COVID-19, while the rest either refused to vaccinate their children (35.6%) or were unsure whether they would (26.1%). Participants’ perception about the effectiveness (OR= 0.726, 95% CI=0.541-0.975, p=0.033) and safety (OR= 0.435, 95% CI=0.330-0.574, p<0.0001) of COVID-19 vaccines were significantly associated with parental refusal for providing the vaccine for their children. Participants who had received or who were planning to take COVID-19 vaccine were sig-nificantly less likely to reject vaccinating their children (OR= 0.156, 95% CI=0.063-0.387, p<0.0001).
Comment:
These sentences could be rephrased to align the reported OR values: Participants’ perception about the effectiveness (OR= 0.726, 95% CI=0.541-0.975, p=0.033) and safety (OR= 0.435, 95% CI=0.330-0.574, p<0.0001) of COVID-19 vaccines were significantly associated with parental refusal for providing the vaccine for their children
-The abstract was modified as follows: “Participants’ perceptions about the effectiveness (OR= 0.726, 95% CI=0.541-0.975, p=0.033) and safety (OR= 0.435, 95% CI=0.330-0.574, p<0.0001) of COVID-19 vaccines were significantly associated with parental acceptance of having their children vaccinated”
Comment:
Conclusion: More efforts should be made to increase parental acceptance of childhood COVID-19 vaccinations in Iraq. -this is rather general and generic statement
-The conclusion was modified as follows: “More efforts including educational and awareness campaigns about the safety and efficacy of COVID-19 vaccines should be made to increase parental acceptance of childhood COVID-19 vaccinations in Iraq.”
INTRODUCTION
Comment:
The aim should be explicitly statement in this section.
-The following was added to the end of the introduction: “The aim of the current study was to evaluate parental acceptance of vaccinating their children and evaluate its associated factors among the Iraqi population”
Materials and Methods
This cross-sectional study used an online survey distributed as a Google Forms link. The questionnaire was distributed to different regions of Iraq using public social networking sites including Facebook, WhatsApp, Viber, Instagram and Telegram. The questionnaire was distributed between May-July 2021.
Comment: any measure taken to prevent duplicated submission from the same person via different channels?
-Unfortunately, duplicated submissions can only be prevented in Google Forms by requesting respondents’ email addresses which may jeopardize the anonymity of the questionnaire, however, visual and computerized (using SPSS tools) were utilized to detect any duplicated responses.
Statistical Analysis
To evaluate internal consistency of the questionnaire, Cronbach’s alphas were computed for the protective practices and knowledge scales. The minimum acceptable Cronbach’s alpha was 0.7.
Comment:
Cronbach’s alpha was used to evaluate the internal consistency of the items used to compute the knowledge and practice scores. The minimum acceptable alpha for the practice score and knowledge estimated using the binary data should be included.
-The following was added: “The minimum acceptable Cronbach’s alpha was 0.7 for the attitude score and 0.5 for the knowledge score as lower Cronbach’s alpha is accepted for binary data [29].”
Reviewer comments:
Comment:
The aim of the current study was to evaluate parental intention to vaccinate their children and the variables associated with it. The study is an extension of the work conducted by the same group of researcher https://www.dovepress.com/covid-19-vaccination-acceptance-and-its-associated-factors-among-the-i-peer-reviewed-fulltext-article-PPA
-Thank you for your comment. This was mentioned in the manuscript in the method section: “The study utilized a previously validated questionnaire used to measure parents' willingness to vaccinate their children against the COVID-19 in Jordan [24]”
Comment:
The manuscript is well written and carefully tabulated results with reliable statistical analysis. I have some minor comments that the authors could consider
ABSTRACTS:
Only 38.3% of the participants were willing to vac-cinate their children against COVID-19, while the rest either refused to vaccinate their children (35.6%) or were unsure whether they would (26.1%). Participants’ perception about the effectiveness (OR= 0.726, 95% CI=0.541-0.975, p=0.033) and safety (OR= 0.435, 95% CI=0.330-0.574, p<0.0001) of COVID-19 vaccines were significantly associated with parental refusal for providing the vaccine for their children. Participants who had received or who were planning to take COVID-19 vaccine were sig-nificantly less likely to reject vaccinating their children (OR= 0.156, 95% CI=0.063-0.387, p<0.0001).
Comment:
These sentences could be rephrased to align the reported OR values: Participants’ perception about the effectiveness (OR= 0.726, 95% CI=0.541-0.975, p=0.033) and safety (OR= 0.435, 95% CI=0.330-0.574, p<0.0001) of COVID-19 vaccines were significantly associated with parental refusal for providing the vaccine for their children
-The abstract was modified as follows: “Participants’ perceptions about the effectiveness (OR= 0.726, 95% CI=0.541-0.975, p=0.033) and safety (OR= 0.435, 95% CI=0.330-0.574, p<0.0001) of COVID-19 vaccines were significantly associated with parental acceptance of having their children vaccinated”
Comment:
Conclusion: More efforts should be made to increase parental acceptance of childhood COVID-19 vaccinations in Iraq. -this is rather general and generic statement
-The conclusion was modified as follows: “More efforts including educational and awareness campaigns about the safety and efficacy of COVID-19 vaccines should be made to increase parental acceptance of childhood COVID-19 vaccinations in Iraq.”
INTRODUCTION
Comment:
The aim should be explicitly statement in this section.
-The following was added to the end of the introduction: “The aim of the current study was to evaluate parental acceptance of vaccinating their children and evaluate its associated factors among the Iraqi population”
Materials and Methods
This cross-sectional study used an online survey distributed as a Google Forms link. The questionnaire was distributed to different regions of Iraq using public social networking sites including Facebook, WhatsApp, Viber, Instagram and Telegram. The questionnaire was distributed between May-July 2021.
Comment: any measure taken to prevent duplicated submission from the same person via different channels?
-Unfortunately, duplicated submissions can only be prevented in Google Forms by requesting respondents’ email addresses which may jeopardize the anonymity of the questionnaire, however, visual and computerized (using SPSS tools) were utilized to detect any duplicated responses.
Statistical Analysis
To evaluate internal consistency of the questionnaire, Cronbach’s alphas were computed for the protective practices and knowledge scales. The minimum acceptable Cronbach’s alpha was 0.7.
Comment:
Cronbach’s alpha was used to evaluate the internal consistency of the items used to compute the knowledge and practice scores. The minimum acceptable alpha for the practice score and knowledge estimated using the binary data should be included.
-The following was added: “The minimum acceptable Cronbach’s alpha was 0.7 for the attitude score and 0.5 for the knowledge score as lower Cronbach’s alpha is accepted for binary data [29].”
Reviewer 2 Report
All my concerns expecially with regard to methodology have been addressed.
Author Response
We are happy that you are satisfied with our responses
Regards
This manuscript is a resubmission of an earlier submission. The following is a list of the peer review reports and author responses from that submission.
Round 1
Reviewer 1 Report
First of all, I would like to thank for the opportunity to review this paper. COVID-19 is an ongoing pandemic that has resulted in global health, economic and social crises. Actually, the vaccination campaign is the first method to counteract the COVID-19 pandemic; however, sufficient vaccination coverage is conditioned by the people’s acceptance of these vaccines. In this context, the paper under review is aimed at evaluate parental intention to vaccinate their children and the relative associated variables.
The subject under study is certainly very important, especially in the historical period we are experiencing. The article presents interesting results but, but it is nevertheless believed that, given the organization of the contents and the description of the same, the manuscript cannot be published in its current form, especially for its local impact and the small sample. I would like to encourage authors to consider several issues to be improved.
Title: it must be improved. It should better highlight the object of the study.
Introduction: The authors should make clearer what is the gap in the literature that is filled with this study. The authors do not frame their study within the vast body of literature that addressed the issue of acceptance of the vaccination in the adult population, before thinking to their sons. What is the possible international contribution of the study to the literature? The objectives should be better explained at the end of the section.
Methods: The survey was conducted using a non-standard questionnaire. The use of an unreliable instrument is a serious and irreversible limitation of the study. The fact that a similar questionnaire has been used in previous surveys is not sufficient. A validation process must be performed to evaluate the tool in a different population. What about face validity and intelligibility? What are the results of the pilot study?
The enrolment procedure must be better specified. How did the authors choose the way to select the sample? This can represent a great bias origin. How did they avoid the selection bias? The author proposed a minimum sample size, but it is not clear what is the reference population? How large is it? Without the numerical identification of the reference population is not clear the meaning of this sample size. The paragraph 2.2. and 2.3 report the same information.
Statistical analysis: I suggest to insert a measure of the magnitude of the effect for the comparisons. Please consider to include effect sizes.
Discussion: I also suggest expanding. Emphasize the contribution of the study to the literature. The discussion must be updated with comparison to other sub populations groups and with one of the principal debated argument in this epidemiological context: the use of a green pass linked to vaccination practice and its impact on vaccination acceptance and hesitancy (refer to articles with DOI: https://doi.org/10.3390/vaccines9111222) a paragraph should be added with a proper reference.
Reviewer 2 Report
Major comment: The study’s aim , method are results are not coherent. The aim of the current study was to evaluate parental intention to vaccinate their children and variables associated with it. But the respondents consisted of participants and their children.
Abstract
The focus of COVID-19 vaccination campaigns has been the adult population, particularly the elderly and those with chronic diseases. However, COVID-19 can also affect children and adolescents, furthermore, targeting this population can accelerate obtaining herd immunity. The aim of the current study was to evaluate parental intention to vaccinate their children and variables associated with it. Method: An online questionnaire was circulated via Iraqi generic Facebook groups to collect Iraqi parental intention towards their children’s vaccinations. Multinomial regression analysis was conducted to evaluate variables associ-ated with parental vaccination acceptance. 491 participants completed the study questionnaire. Only 38.3% of the participants were willing to vaccinate their children against COVID-19, while the rest either refused to vaccinate their children (35.6%) or were unsure whether they would (26.1%). Participants’ perception about the effectiveness (OR= 0.74, 95% CI=0.55-0.99, P<0.05) and safety (OR= 0.43, 95% CI=0.33-0.57, P<0.01) of COVID-19 vaccines were significantly associated with parental refusal for providing the vaccine for their children. Participants who had received or who were planning to take COVID-19 vaccine were significantly less likely to reject vaccinating their children (OR= 0.163, 95% CI=0.07-0.40, P<0.01). More efforts should be made to increase parental acceptance of childhood COVID-19 vaccinations in Iraq.
2.2. Study design and subjects
The required number of participants for this study was calculated at 95% confidence 87 interval and 5% confidence level and was equal to 384 participants. The Rule of Events 88 per Variable criterion (EPV) ≥10 13 was used to determine the required sample to carry 89 out a logistic regression that estimated variables association with participants' reluctance 90 to give their children vaccine. The number of independent variables that can be included 91 in the current model is 13. The logistic regression model included 11 variables.
Comment: 13?
2.3. Sample size calculation
The required number of participants for this study was calculated at 95% confidence interval and 5% confidence level and was equal to 384 participants. The Rule of Events per Variable criterion (EPV) ≥10 [13] was used to determine the required sample to carry 96 out a logistic regression.
Comment: repeated info
Table 1. Sociodemographic characteristics of the participants and their children.
|
Marital status |
Single |
105 (21.4) |
|
|
Married |
386 (78.6) |
||
Reviewer 3 Report
- Some edits are needed such:
- -L10, L37: spell the COVID-19 first
- L23: Delete "Method"
- -L25 do not start with number
- L29,32: Provide the exact p-value. use p not P
- L28-32: Be consistent with OR value. Three decimals.
- L43 24 thousand
- and soon. Please proofread again
- L71: Explain more about: how the link was distributed in various Facebook Iraqi generic groups.
- L77-80: How authors excluded these group where many parents do not know whether the children hace cancers for example.
- L87: Please give justification. I am not convince only 384 is required for one country.
- L87-92: These are not part of "Study design and subjects". Please change
- 2.2 and 2.3 are repetitive
- L102: Who were the participants and how they were selected randomly?
- L134: Please provide Cronbach’s alpha for both domains from this study
- L139: Pleas include the data that were excluded. What is the response rate? Possible to identify?
- L151-152: Please revise
- L160-165 and the entire manuscript: Please provide the exact p-value. Use p not P. p-value should be three decimals for example 0.071 not 0.07. Please provide p<0.001 not <0.01.
- Table 2:
Are you working/studying in a medical field? (Yes)
Do you have a Chronic disease? (yes)
Child risk towards COVID-19 (high risk)
Have you taken or planning to take COVID-19 vac-cine? (Yes)
Both options should be provided such Yes and Not.
Table 3: For the first four characteristics: How the authors assessed them. What scale are they (ratio?) These are not clear. There are no information in Method12. L214-156: Please edit this to enhance the discussion because this no only true for the parents but also in general population. Please add the sentence: "This is also one of the barriers to accept the COVID-19 vaccine in general population (Ref https://doi.org/10.52225/narra.v1i3.57 and https://doi.org/10.52225/narra.v1i3.55.
13. L278: The sample size is one of the limitation of this study and this statement is contradict.
14. Please discuss about the selection bias clearly in this study.
15. L284: Please delete